# Reframing the Brain Age Prediction Problem to a More Interpretable and Quantitative Approach

**Neha Gianchandani** [1 2]  **Mahsa Dibaji** [3]  **Mariana Bento** [1 2 3 4]  **Ethan MacDonald** [1 2 3 4 5]  **Roberto Souza** [2 3]

## Abstract

Deep learning models have achieved state-of-the-art results in estimating brain age, which is an important brain health biomarker, from magnetic resonance (MR) images. However, most of these models only provide a global age prediction, and rely on techniques, such as saliency maps to interpret their results. These saliency maps highlight regions in the input image that were significant for the model's predictions, but they are hard to be interpreted, and saliency map values are not directly comparable across different samples. In this work, we reframe the age prediction problem from MR images to an image-to-image regression problem where we estimate the brain age for each brain voxel in MR images. We compare voxel-wise age prediction models against global age prediction models and their corresponding saliency maps. The results indicate that voxel-wise age prediction models are more interpretable, since they provide spatial information about the brain aging process, and they benefit from being quantitative.

## 1. Introduction

Researchers have hypothesized that the brain age of healthy subjects should match their corresponding chronological ages. Increased brain age compared to chronological age is an important indicator of brain health, making brain age prediction a widely explored research area. Most brain age prediction work focuses on predicting a 'global' brain age index that reflects the maturity level and age of the brain.

The global brain age index has been shown to be an effective biomarker to assess the aging process as well as to understand structural changes in the brain in the presence of neurological disorders (Cole & Franke, 2017; Wang et al., 2019). Global brain age has most widely been estimated from T1-weighted Magnetic Resonance (MR) volumes using Deep Learning (DL) techniques (Cole & Franke, 2017; Tanveer et al., 2023; Jónsson et al., 2019), as a regression task.

Predicting the brain age is important to study normal brain aging and prospective cognitive decline, but understanding and interpreting these findings, and uncovering the black-box nature of these methods is an even more relevant task. These models aim to detect brain regions or biomarkers that should be analyzed in greater detail, leading to personalized diagnosis and decision-making. Efforts have been made to interpret the DL models with techniques based on saliency maps, such as Gradient-weighted Class Activation Mapping (Grad-CAM) (Bermudez et al., 2019; Yin et al., 2023; Selvaraju et al., 2017), and occlusion-based techniques (Bintsi et al., 2021; Zeiler & Fergus, 2014). Saliency map techniques focus on creating visualizations to depict the contribution levels of each pixel in the decision-making process, whereas occlusion-based techniques aim to identify regions that are most important to make a particular decision by hiding regions in the input and observing their impact on model performance. The worse the model performs when hiding a certain feature, the higher its importance. There are other interpretability techniques like Layer Wise Relevance Propagation (Bach et al., 2015) and SHapley Additive exPlanations (SHAP) (Lundberg & Lee, 2017), however, to the best of our knowledge, they have not been utilized to explain brain age prediction models.

At a high level, the aforementioned techniques provide an understanding of how DL models learn and explain the relationship between the features and the models' decision making, hence, providing interpretability to DL models, *i.e.*, unboxing the black-box. Existing interpretability techniques help with understanding the global brain age prediction models by accessing the DL model's decision-making. Such techniques explain what features in the model input contribute to the decision-making process (*i.e.* to predict global

[1]Department of Biomedical Engineering, University of Calgary, Calgary, Alberta, Canada [2]Hotchkiss Brain Institute, University of Calgary, Calgary, Alberta, Canada [3]Department of Electrical and Software Engineering, University of Calgary, Calgary, Alberta, Canada [4]Alberta Children's Hospital Research Institute, Calgary, Alberta, Canada [5]Department of Radiology, University of Calgary, Calgary, Alberta, Canada. Correspondence to: Neha Gianchandani <neha.gianchandani@ucalgary.ca>.

*Workshop on Interpretable ML in Healthcare at International Conference on Machine Learning (ICML)*, Honolulu, Hawaii, USA. 2023. Copyright 2023 by the author(s).

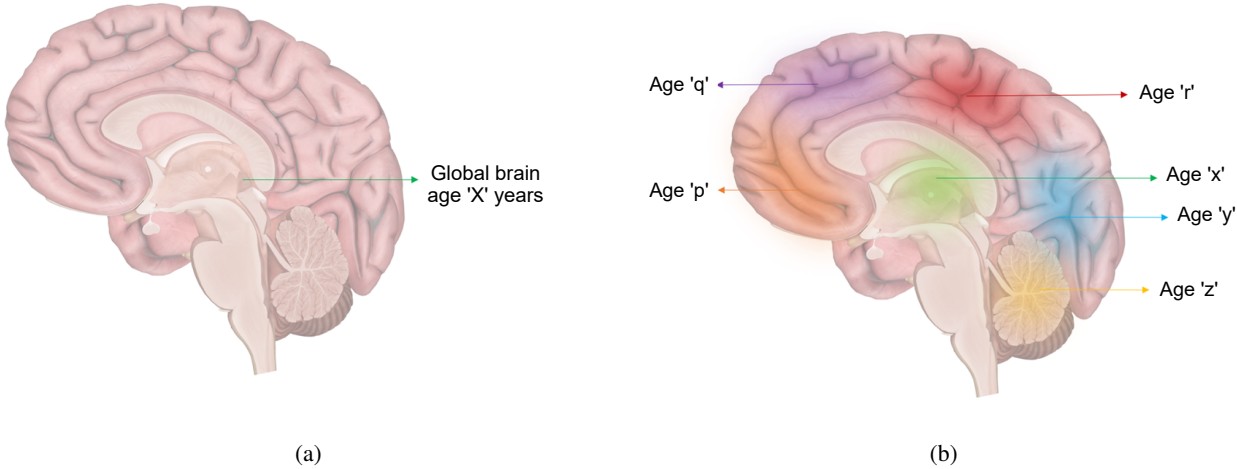

*Figure 1.* (a) Traditional paradigm where a brain age prediction model is used to predict a global brain age value for the whole brain. (b) New proposed paradigm where a voxel-level brain age prediction model assigns different brain ages to each region of the brain. At the most granular level, each voxel corresponding to brain tissue in the image can be assigned a brain age.

brain age) providing a spatial-level analysis of the problem, relevant to identify specific brain regions of interest, biomarkers, and abnormalities related to aging.

It is important to highlight that most saliency-based interpretability techniques were initially proposed to explain classification models, producing heat maps to a specific class. The translation of such techniques for regression tasks is not straightforward.

In this article, we propose a brain age prediction model with improved interpretability. Interpretability has been defined differently in varying contexts (Doshi-Velez & Kim, 2017; Kim et al., 2016; Miller, 2019), however, all definitions aim to achieve a common goal: to satisfy human curiosity (Miller, 2019) and in the machine learning context, to make the modeling pipeline (from feature extraction to decision making) less of a black box, and more transparent. An alternative to utilizing interpretability methods to explain the feature extraction by previously trained age prediction models, we propose to redefine how we approach the research problem. The aim of predicting brain age is to understand how the brain ages in healthy compared to diseased brains. We further want to understand how various regions contribute to the decision-making process reflecting more on the spatial aging processes in the brain. Instead of approaching it as a global age prediction problem, we propose to approach it as an image-to-image regression problem where we predict brain age at a voxel level. So far to the best of our knowledge, only (Popescu et al., 2021) have attempted a voxel-level brain age prediction approach. The authors use a U-Net-like (Ronneberger et al., 2015) architecture to predict voxel-level brain age and achieve a Mean Absolute Error (MAE) of $9.94 \pm 1.73$ years. However, the approach

is not discussed from an interpretability point of view.

A voxel-level approach will enable a spatial-level analysis of the brain aging process by assigning a brain age prediction to each voxel of the brain. Individual predictions for each volume unit not only allow us to analyze the aging process at a more fine-grained level but also provides a quantitative visualization that reflects how different regions of the brain are aging. A brain voxel in the image that is assigned an increased or decreased brain age index can be explained by the contribution of the voxel in the decision-making process by the DL model. In this article, we propose a voxel-level age prediction model as a new approach to interpretability. An overview of global age versus voxel-level age prediction outputs is described in Fig. 1.

We further compare our new proposed method to the existing methods for interpretability. Grad-CAM (Selvaraju et al., 2017), Occlusion Sensitivity maps (Zeiler & Fergus, 2014) and SmoothGrad (Smilkov et al., 2017) are utilized to explain a publicly available state-of-the-art global brain age prediction model, and the interpretations are discussed and compared with our proposed voxel-level brain age prediction model.

We acknowledge the existing state-of-the-art methods for brain age prediction and emphasize that the focus of this article is not to propose state-of-the-art brain age prediction models, but to contrast existing interpretability techniques with our proposed approach to the brain age prediction problem to uncover the black box and to better understand the predictions.

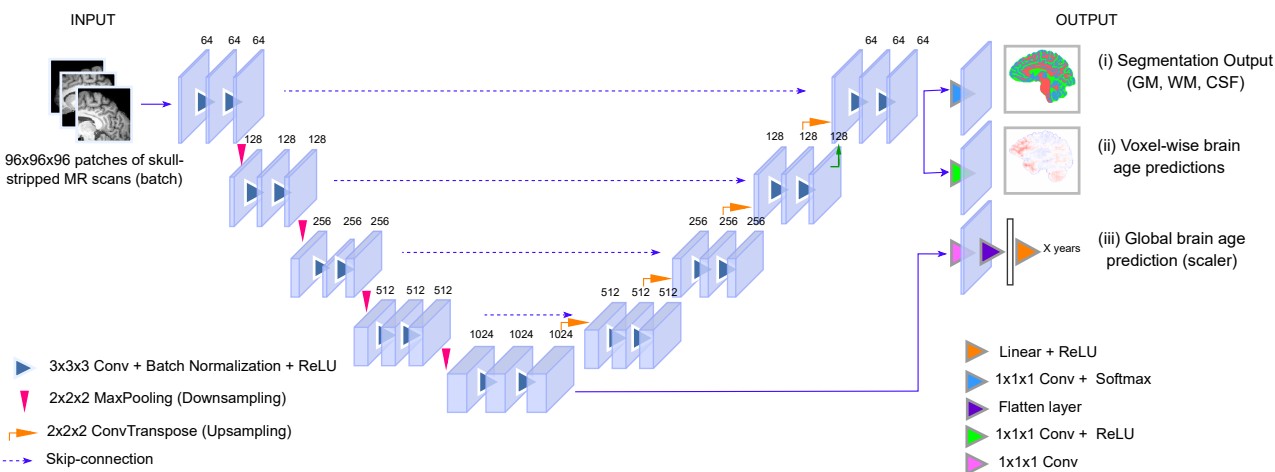

*Figure 2.* Proposed voxel-level brain age prediction model architecture. The model has a U-Net backbone and follows a multi-output design with three outputs: (i) Segmentation of Gray Matter, White Matter and CSF, (ii) Voxel-level brain age prediction, and (iii) Global-level brain age prediction.

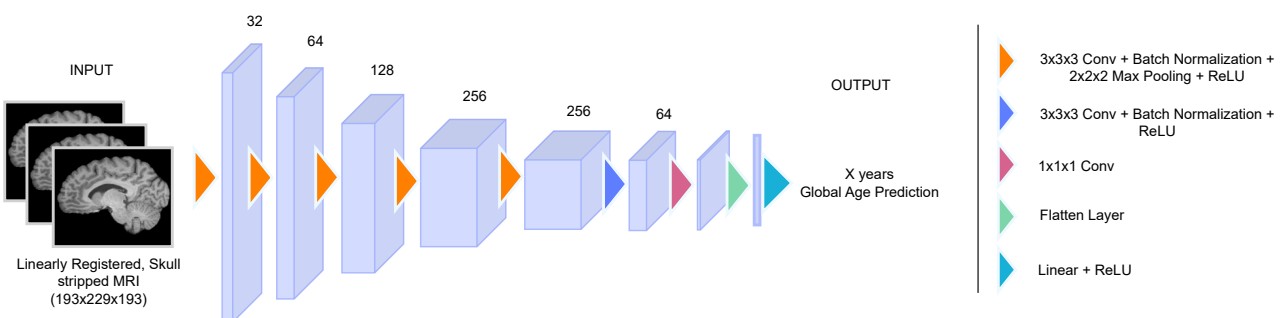

*Figure 3.* Global age prediction model architecture. The backbone is adopted from (Peng et al., 2021), and the output head is modified to treat global age prediction as a regression problem rather than a classification problem as done in the original research.

## 2. Materials and Methods

### 2.1. Data

We used 3D T1-weighted MR scans from the publicly available Calgary-Campinas dataset (Souza et al., 2018). We will refer to this dataset as $D_{cc}$ hereafter in this article. All data corresponds to presumed healthy subjects. $D_{cc}$ has 359 samples aged 29-80 years (mean age of 53.47±7.84) with a male:female sex ratio of 49:51 percent. The data was acquired on three different scanners (Philips, Siemens, and General Electric), each at two magnetic field strengths (1.5 T and 3 T). Skull-stripping masks, as well as tissue segmentation masks (Gray Matter (GM), White Matter (WM), and Cerebrospinal Fluid (CSF)), are also publicly available with the dataset.

### 2.2. Data Preprocessing

For the voxel-level brain age prediction model, the scans are reoriented to a standard template (MNI152 (Fonov et al., 2009)) using the FSL (Jenkinson et al., 2012) utility 're-orient2std' to ensure consistency across the dataset. We adjusted each sample's intensity to fall within the range of 0 to 1. Any other kind of preprocessing steps, such as registration is avoided for this model as the predictions are to be done at a voxel level, and we want to ensure that the input features remain untouched and the brain structures hold the shape and intensity values as in the original reconstructed T1-weighted image.

For the global age prediction model, the MR scans are linearly registered with 6 degrees of freedom using FMRIB Software Library's (FSL) FLIRT (FMRIB's Linear Image

Registration Tool) command (Jenkinson & Smith, 2001), which allows for rotation and translation, keeping the shape of the brain consistent to avoid loss/change in input data features. The registration step was seen to improve the performance of the global age prediction model and hence, was included in the pipeline.

## 2.3. Proposed Model Architectures

### 2.3.1. VOXEL-LEVEL BRAIN AGE PREDICTION MODEL

The proposed model follows the 3D U-Net (Ronneberger et al., 2015) architecture backbone (Fig. 2). The model has an encoder network (left) and decoder network (right) joined together with a bottleneck (center bottom), making an U shape. The input is downsampled at each level of the encoder as features are learned, whereas the feature map is interpolated back to the original input size iteratively at each level of the decoder. Skip connections are used to connect the encoder and the decoder at each level to allow for feature re-usability as well as help with the vanishing gradient problem. Batch normalization is used post convolution layers in the encoder network.

The model follows a multitask architecture with three outputs for which the features are learnt simultaneously. This forces the model to learn similar features for the three tasks. The three tasks defined for the model are: (i) A segmentation task to segment GM, WM, CSF. (ii) Voxel-level brain age prediction task. The U-Net architecture helps in maintaining output size to be the same as the input so as to obtain voxel-level brain age prediction for each voxel in the input age. A Rectified Linear Unit (ReLU) activation is used to ensure positive age predictions. (iii) Global-level brain age prediction task, which is computed from the bottleneck features of the model and is a scalar age prediction for each volume. This task closely resembles the output from the global age prediction model described in Section 2.3.2.

Task (i) and (iii) are helper tasks that aid the model in learning accurate features for the voxel-level age prediction task. Adding the segmentation task is especially useful as it pushes the model to learn structural features for the voxel-level age prediction task ignoring other unnecessary information present in the input images. The global-level brain age can be thought of as a prerequisite and simpler version of the voxel-level brain age prediction task.

### 2.3.2. GLOBAL BRAIN AGE PREDICTION MODEL

We adapt the architecture for the global brain age prediction model from a state-of-the-art Simple Fully Convolutional Neural network (SFCN) proposed in (Peng et al., 2021; Gong et al., 2021). The authors treat the brain age prediction task as a soft classification task where the model predicts a Gaussian Probability distribution centered at the ground truth (chronological age) instead of a scalar brain age index. The model is lightweight as it is made up of 7 convolution blocks where the input is down-sampled after each convolution layer in the first five blocks with $3 \times 3 \times 3$ convolutions, followed by a $1 \times 1 \times 1$ convolution block and a classification head. Batch normalization is also used to ensure a smooth training process. We modify the architecture and retain the convolution blocks, but treat it as a classical regression problem (Fig. 3).

## 2.4. Loss Function

### 2.4.1. VOXEL-LEVEL BRAIN AGE PREDICTION MODEL

A combination loss function is used to train the voxel-level brain age prediction model. The loss function is a weighted sum of three loss terms, one corresponding to each task. The Soft Dice Score (Milletari et al., 2016) is used for the segmentation task, and MAE at the global and voxel level is used for the brain age prediction task.

The weighting terms ($\alpha$, $\beta$, and $\gamma$) used are initialized and changed as the model trains in a way that all tasks are given equal importance throughout the training process. The loss function is described in equation 1.

$$L_{overall} = \alpha Dice_{loss} + \beta MAE_{global} + \\ \gamma MAE_{voxel} \tag{1}$$

### 2.4.2. GLOBAL BRAIN AGE PREDICTION MODEL

We use the MAE as the loss function for the global age prediction model. The decision to use MAE as the loss function was reinforced after running experiments with Mean Squared Error (MSE) as the loss function. Experiments showed that MAE performed significantly better leading to an improved training trajectory with a lower validation loss and a smoother training/validation curve.

## 2.5. Training Methodology

### 2.5.1. VOXEL-LEVEL BRAIN AGE PREDICTION MODEL

We train the model for 300 epochs with an initial learning rate of 0.001, which reduces every 70 epochs by a multiplicative factor of 0.5. A batch size of 8 is used for training where each input sample is a randomly cropped patch of size $96 \times 96 \times 96$ from the T1-weighted volume. A single crop is randomly obtained from each volume ensuring significant brain volume in each crop while exposing the model to the same brain region from various perspectives. 50% of the cropped patches are rotated by $15°$ as an augmentation step on-the-fly. To prevent the model from solely learning to predict global brain age for each voxel, we introduce small noise to the ground-truth labels before calculating the loss.

For voxel-level age prediction task, instead of the ground truths having the the same global age at each voxel, we introduce small noise (in the range of [-2,2]) to ensure the models learns variations in age across different voxels (or regions) without significantly impacting the error.

### 2.5.2. GLOBAL BRAIN AGE PREDICTION MODEL

In the training process for the global brain age prediction model, we ran the model for 50 epochs, starting with a learning rate of 0.0001. As the training progressed, learning rate was decreased by half every 20 epochs. We trained with a batch size of 8, and adjusted each sample's intensity to fall within a scale of 0 to 1. We also implemented an on-the-fly augmentation step, rotating 50% of the samples by $15°$. This approach aimed to increase the model's robustness by providing a diverse set of sample variations throughout the training process.

### 2.6. Interpretabily Methods

There are several techniques employed to explain the decision-making process of a deep-learning model, such as Grad-CAM (Selvaraju et al., 2017). This method uses the gradients flowing into the final convolutional layer to produce a coarse localization map, highlighting the image regions important to prediction. Its model-agnostic property makes it applicable across a wide range of Convolutional Neural Network (CNN)-based models, without requiring retraining. However, the main limitation of Grad-CAM is its coarse localization due to the low spatial resolution of deeper convolutional layers, which can sometimes limit its interpretability. To counteract this limitation, we employed a method that averages two maps—one from an earlier convolutional layer as the target layer and one from the final layer—to blend detailed features with high-level representations for a more comprehensive understanding (Morbidelli et al., 2020). It is important to acknowledge that the inputs to the two layers considered for generating the heatmaps are different in terms of the input feature maps as well as dimensions. Aggregating two feature maps, originating from layers at distinct depths in the model helps to smooth out inconsistencies or noise that may exist in the heatmap from the final convolutional layer. It also helps in obtaining informative heatmaps by encompassing both relatively high-level and low-level features present at different depths in the network (McAllister et al., 2020).

Another interpretability technique, Occlusion Sensitivity (Zeiler & Fergus, 2014), provides an interpretability approach that systematically occludes portions of the input image to observe changes in model output, thereby producing a heat map of the input's most influential regions. The strength of this technique lies in its simplicity and general applicability; it can be applied to any model predicting based on an image input, even black box models, without requiring access to model gradients. Nevertheless, the technique is not without drawbacks. Specifically, Occlusion Sensitivity is computationally intensive, necessitating a rerun of the model for every occluded version of the image, which makes it somewhat inefficient for larger images or complex models. Additionally, the results can be influenced by the size and shape of the occluding patch, making it important to choose these parameters carefully.

Finally, SmoothGrad (Smilkov et al., 2017) offers another approach to interpretability. In contrast to the previous techniques, SmoothGrad aims to reduce noise in gradient-based saliency maps by averaging the gradients of multiple noisy versions of the same input image. Consequently, it often yields less noisy saliency maps than single input-based methods, highlighting noise-robust patterns. However, Smooth-Grad requires multiple forward passes through the model as well as repeated gradient calculation to obtain smoother saliency maps for each input, which increases computational demands. Similarly to Occlusion Sensitivity, the choice of appropriate noise level may prove challenging and require further experimentation or tuning.

For our experiments, we implement interpretability methods using MONAI (Cardoso et al., 2022), which is a PyTorch-based framework that provides built-in functions for implementing various interpretability techniques.

## 3. Results

The voxel-level brain age prediction model achieved an MAE of $5.96\pm3.75$ years on the $D_{cc}$ test set (n=30 samples). The global brain age prediction model achieved an MAE of $6.56 \pm 4.04$ years on the same test samples (see Table 1).

*Table 1.* Results for voxel-level and global brain age prediction models on the $D_{cc}$ test set

| MODEL | TEST SET | MAE±S.D |
|---|---|---|
| VOXEL-LEVEL MODEL | $D_{cc}$ | $5.96\pm 3.75$ |
| GLOBAL MODEL | $D_{cc}$ | $6.56\pm 4.04$ |

For the voxel-level age prediction model, we present results using MAE averaged across all voxels of the brain for simplicity. It is not feasible to report voxel-level MAE for all (millions) of the brain voxels individually. However, for visualization, we use predicted age difference (PAD) masks that show the difference between the predicted brain age and the chronological age at the level of each voxel. Blue color indicates brain regions that look younger than chronological age and red points to older-looking brain regions. PAD masks can be an excellent way to visualize brain maturity levels from voxel-level brain age predictions. The masks also provide an additional level of quantification to explain

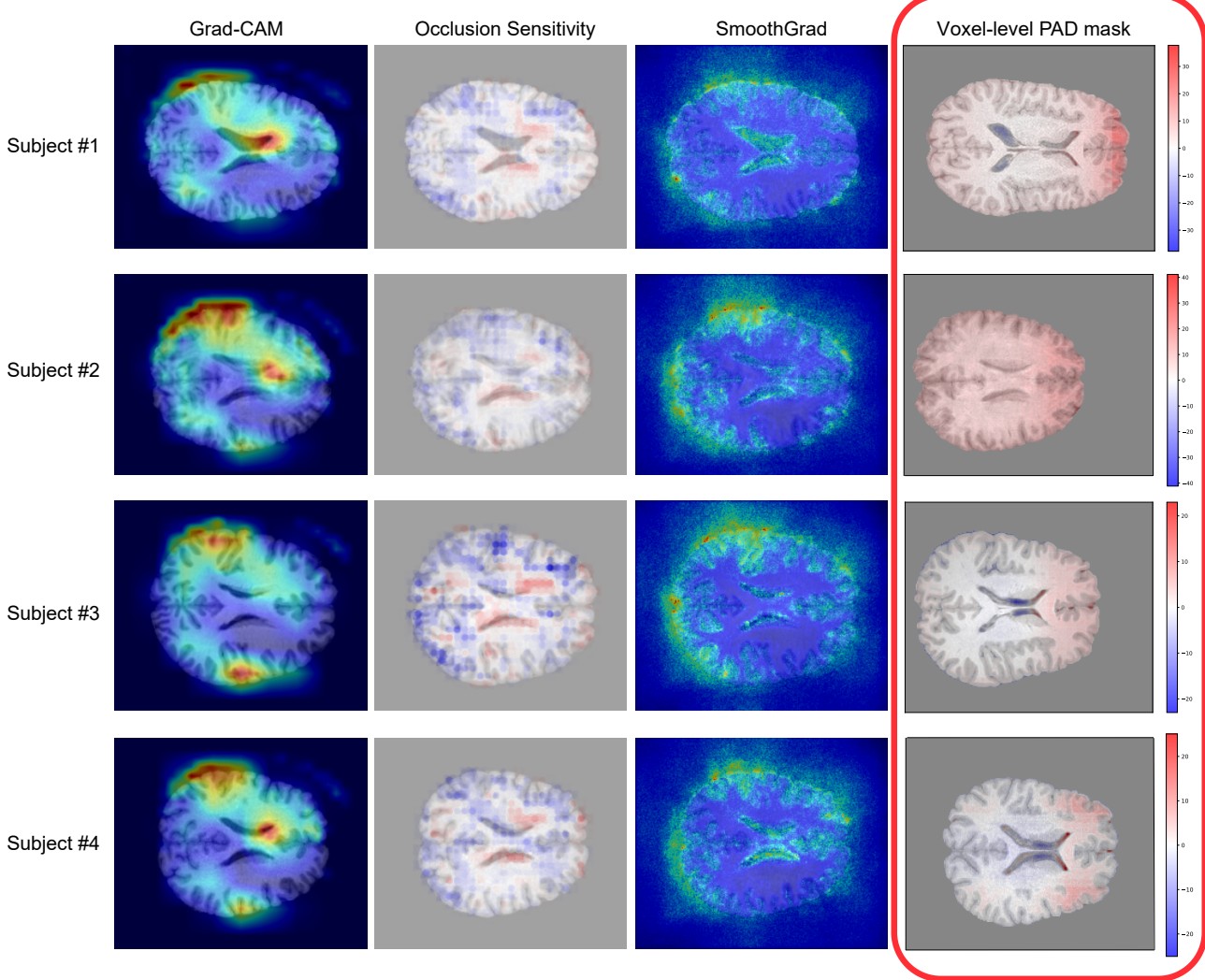

*Figure 4.* Traditional interpretability methods (left to right- Grad-CAM, Occlusion Sensitivity maps, and SmoothGrad) implemented on global age prediction model and voxel-level brain PAD masks obtained from voxel-level brain age prediction model.

the outputs of the DL model.

For the global age prediction model, we report the MAE over the test set and also present the results of three different interpretability techniques (Fig. 4).

## 4. Discussion

Figure 4 shows the comparison of traditional interpretability methods to voxel-level PAD masks. Each row represents different interpretability outputs for the same test subject. Grad-CAM heatmaps show the important regions that contributed to the decision-making process (to predict global brain age) using a red-yellow-blue gradient colormap with red regions being the biggest contributors and blue regions

being the smallest contributors or least important. However, as can be observed in Fig 4, Grad-CAM heatmaps are coarse and prone to up-sampling errors. Grad-CAM maps are also not comparable across different samples as they are hard to quantify, this is due to the origin of the heatmap intensity values which come from the relative strength of gradients for a particular input image.

Similarly, the occlusion sensitivity maps for classification problems show the most and least important regions for classifying a specific class, however, in the context of a regression problem such as brain age prediction, red points refer to regions that make the model overestimate the brain age (prediction > ground truth) and blue points refer to regions that make the model underestimate brain age (pre-

diction < ground truth). This means that all regions in red and blue contribute to the brain age prediction decision in some way, and the white regions are the least contributing.

Saliency maps created using SmoothGrad highlight the areas that significantly influence the model's output. These crucial regions are typically marked in red, while the areas having a minimal effect are illustrated in blue. As observed, these maps offer more detailed insights compared to Grad-CAM, seemingly highlighting important regions with more precision. This enhanced level of detail could be attributed to the fact that SmoothGrad averages over multiple versions of the same input with added noise. Similar to Grad-CAM, SmoothGrad is also a gradient-based technique, it relies on the quantitative nature of the gradients, and how changing the input affects the gradients in the model. Thus, it reflects the relative importance of different regions in the input image (Brain MR).

Overall, while distinct differences exist in the maps generated by the three techniques, a general agreement can be seen regarding the key regions utilized in predicting global age. While the traditional interpretability methods do a good job of providing a high-level understanding of important regions, they are based on relative scores for a specific input sample, and not absolute quantitative measurements which can be used to directly compare different samples.

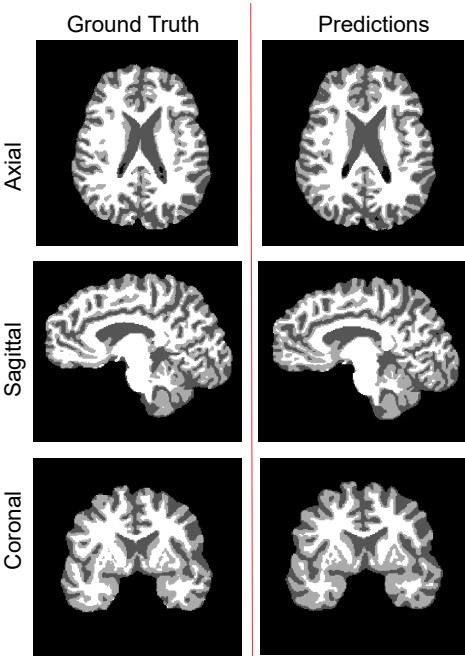

| Ground Truth | Predictions |
|---|---|

*Figure 5.* Segmentation results using the voxel-level brain age prediction model on one sample. The model achieved a mean dice score of 84.25% on the test set.

The voxel-level PAD masks on the other hand are used to visualize individual predictions for each voxel (region) of the brain. The PAD values embedded in the masks quantify the difference between the brain age prediction and chronological age (in years), consequently making the PAD masks comparable across different samples. Owing to the multitask design of the model architecture, the model is forced to learn features that can be repurposed for all three tasks. The addition of the brain tissue segmentation task ensures that the model learns structural features within the brain region that will be used for the segmentation as well as the voxel-level brain age prediction task. We achieve a Dice Score of 84.25% on the $D_{cc}$ test set indicating a high overlap between predicted segmentation and ground truth for GM, WM, and CSF, hence accurate segmentations (Fig. 5). This is only possible if correct structural features are learned during the feature extraction process while training the model. This further confirms that structural features within the brain region are contributing to the voxel-level age prediction task. It can also be hypothesized that big PAD values can be partially explained by the presence of underlying structural anomalies that have not evolved into a disease suggesting the contribution of the region in predicted voxel-level brain age. Fig. 4 shows PAD masks of healthy subjects from our test set, research has shown that the brain aging process varies across different regions of the brain. Studies have also shown that each healthy brain is unique and follows a different spatial aging trajectory (Raz et al., 2005; Scahill et al., 2003) and this can be observed in the PAD masks. The underlying structural features of the brain cause the predicted age difference of a voxel to be non-zero PAD and this varies spatially in the brain.

Voxel-level PAD maps also provide an increased resolution at the level of each voxel, unlike traditional interpretability methods which are usually noisy at such granular level (Liu et al., 2021). Saliency map-based techniques like Grad-CAM and even SmoothGrad only indicate the activity of broad regions in the final output, however, due to the heatmaps undergoing interpolation (for upsampling), it becomes nonviable to correlate the importance of regions to specific morphological changes that have caused a particular prediction. Age prediction at a voxel level can lead to a fine-grained analysis of underlying changes in the brain. For the purpose of this article, our aim is to compare how various traditional interpretability methods compare to voxel-level brain PAD masks, hence, we will base all our comparisons on the voxel-level PAD maps. However, in the future, to correlate variation observed in PAD maps to existing research on aging, which often focuses on regions rather than individual voxels, it can be valuable to average the voxel-level brain age predictions within the known anatomical regions of the brain to visualize and study the variation in aging trajectories across regions of the brain. In the future, it will

also be valuable to get clinical feedback to better understand the aging patterns observed in the voxel-PAD maps. This can help correlate the spatial variations observed in the brain to existing research on aging at a regional level.

Occlusion-based sensitivity maps come close to providing similar insights as voxel-level PAD maps, however, with occlusion sensitivity maps, we only get an estimation of single regions contributing to over or under-estimation of overall global brain age. The impact of an occluded region on the output is hard to quantify as it's rarely ever one region in the image that contributes entirely to the final output. The output of DL models is a combination of multiple features, and occlusion sensitivity maps do not account for the impact of occluding multiple regions together.

Most traditional interpretability techniques have the advantage of being utilized for multiple use cases and with different model architectures, however, they are used as a post-modeling step to assess if the trained model has learned accurate features. Our proposed model on the other hand is specific to the brain age prediction problem (as of now, although, can be extended to other imaging research problems), however, it is a modeling technique, rather than an additional algorithm to check for interpretability. It ensures that accurate features are learned to predict brain age and additionally, also provides spatial information on the brain aging process.

## 5. Conclusion

In this article, we propose voxel-level brain age prediction, a step towards interpretability in the brain age prediction realm. This perspective on brain age prediction is relatively new and has not been explored from an interpretability point of view in the past. We also compare the outputs of the proposed voxel-level age prediction model to existing traditional interpretability methods and reflect on the differences between them. Through our findings, we show that our proposed model provides an additional level of interpretability, and fine-grained analysis of the features used for the decision-making process by the model and is quantitative in nature.

## Acknowledgements

The authors would like to express our appreciation to the reviewers for their feedback which helped in refining and strengthening the final version of the paper. This project is supported by the Hotchkiss Brain Institute, University of Calgary, Alberta Innovates, the Natural Sciences and Engineering Research Council of Canada (NSERC), and the Digital Research Alliance of Canada. The authors would like to thank Denvr Dataworks for not only providing access to their high-performance supercluster but also for making our compute-intensive project more sustainable. With their environmentally conscious cloud infrastructure and user-friendly AI Cloud software, we were able to train our models efficiently while conserving water and reducing our carbon footprint. This collaboration allowed us to advance our research in interpretable medical machine intelligence, creating a positive impact on healthcare while demonstrating our commitment to sustainable practices.

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
