# OpenReview forum: "Reframing the Brain Age Prediction Problem to a More Interpretable and Quantitative Approach"
_ICML.cc/2023/Workshop/IMLH — IMLH 2023 Poster_

### Official Review · Reviewer_FRnt · 2023-06-10
**Limited experiment validation (8-page)**

**Rating:** 5
**Confidence:** 3

**Review:**

**Summary:**

This study reframes the problem by estimating brain age for each voxel in MR images, comparing voxel-wise models to global models. Preliminary results suggest that voxel-wise models are more interpretable, providing spatial information and benefiting from quantitative analysis.

**Strength:**

Preliminary results suggest that voxel-wise models are more interpretable and have lower errors.

**Weakness:**

1. The experiment evaluation is limited. As admitted by the authors, this is just preliminary results. There is only one baseline model included in Table 1. Different components of the proposed framework have not been ablatively studied. Without knowing the sensitivity	of different components (e.g., segmentation, voxel-wise branch) with empirical experiments, it is hard for readers to have deep understanding of the model.

2. Figure 1 (b) (as well as other related parts) is confusing for me. Do authors assigned different ages for different regions in a single brain? To me, this is not a realistic setting.

3. The writing and presenting of the paper can be improved (e.g., line 097 lacks period).

---

### Official Review · Reviewer_hf2z · 2023-06-18
**This work proposes a reframing for the age prediction problem from MRI images to an image-to-image regression problem as opposed to the global age prediction model.**

**Rating:** 6
**Confidence:** 5

**Review:**

The proposed model estimates the brain age for each brain voxel in MR images. So, it keeps the spatial information. The corresponding saliency maps from both global prediction and the voxel-wise prediction are then compared. The proposed approach allows analysis of the aging process at a more fine-grained level and also provides a quantitative visualization that reflects how different regions of the brain are aging.

1. The paper is clearly written and easy to follow.
2. Utilizing helper tasks such as segmentation and global age prediction for voxel-level prediction is very intuitive.
3. While a voxel-level brain age prediction seems novel, the question may arise if the brain age is correlated with spatial interaction among regions rather than independently related with voxels.
4. How the ground-truth (chronological) age for each voxel has been used during training is not clear. But it's reasonable to assume each of the voxels may have different characteristics because they are not tied to chronological age only.
5. While it’s obvious that not all saliency-based techniques are applicable to regression setting, relevance-based maps or Shapley values are not inconsistent for regression models. It might not be known to the authors that SHAP and LRP have been used to explain brain age prediction models, e.g.:
*SHAP: Ball, Gareth, et al. "Individual variation underlying brain age estimates in typical development." Neuroimage 235 (2021): 118036.*
*LRP: Hofmann, Simon M., et al. "Towards the interpretability of deep learning models for multi-modal neuroimaging: Finding structural changes of the ageing brain." NeuroImage 261 (2022): 119504*
6. It seems the study is using two global age prediction models: 1) a global age prediction model with the voxel-level prediction as a multitask model 2) a separate global age prediction model adapted from the previous work. Which global age prediction model was interpreted using traditional interpretability methods?
7. The study proposed a solution for low resolution of Grad-CAM maps by averaging two maps—one from an earlier convolutional layer as the target layer and one from the final layer. It may resolve the problem to some extent. However, it is still not able to identify the fine-details in the input space. Also, averaging two maps (one possibly upsampled I guess) may confuse the details because they have their respective resolutions and work on very different inputs.
8. While authors have already mentioned some pitfalls, occlusion sensitivity also has its inherent problem of missing correlations among regions.
9. Explaining PAD values may be misleading and have human confirmation bias. Authors have mentioned that studies have shown that each healthy brain is unique and follows a different spatial aging trajectory. So, how are these PAD values interpretable?
10. Overall, the paper is worth for discussion in the neuroimaging (healthcare) community.

---

### Official Review · Reviewer_qWc4 · 2023-06-18
**Solid interpretable analysis of real-world clinical problem**

**Rating:** 6
**Confidence:** 3

**Review:**

The authors study the problem of brain age prediction using MRIs. Focusing on the interpretability aspect, they propose a multi-task model that predicts age at each voxel, along with brain tissue segmentation and a global age prediction. They compare the heatmaps generated by the voxel level prediction error against other interpretability methods.

The paper presents a solid interpretable approach to tackle a real-world clinical problem, and would be a good contribution to the workshop.

I have the following comments and suggestions:
1. The authors mention the importance of their multi-task approach multiples times. They should conduct ablation studies on each of the tasks to indeed demonstrate its importance (e.g. what happens to the voxel heatmaps if the segmentation task is removed?)
2. I am not convinced that heatmaps generated by the author's method is better than the baseline methods. The paper would be greatly strengthened by a real-world user study with clinicians.
3. The authors should explore how good the voxel level age predictions actually are, as this is a critical part of their visualization. Are there regions/voxels that consistently have the lowest prediction error? How does an aggregated voxel-level age prediction compare with the global age prediction from the multitask model?

---

### Meta-Review · Area_Chair_vddN · 2023-06-19

**Recommendation:** Accept (Poster)
**Confidence:** 4

**Metareview:**

This work proposes a reframing for the age prediction problem from MRI images to an image-to-image regression problem as opposed to the global age prediction model.

All reviewers appreciated the relevance to the workshop. One reviewer raised concerns that the evaluations were not comprehensive and require more validations. Other reviewers also suggested a few other details including relevant baselines and more discussions on specific choices.

Overall, the paper is recognized to be worth discussing in the workshop. The AC recommends acceptance.

---

### Decision · Program_Chairs · 2023-06-20

Accept (Poster)